# *MUC5B* rs35705950 Promoter Variant Is Associated with Usual Interstitial Pneumonia in Patients with Antisynthetase Syndrome

**DOI:** 10.3390/jcm13206159

**Published:** 2024-10-16

**Authors:** Daphne Rivero-Gallegos, Mayra Mejía, Karol J. Nava-Quiroz, Espiridión Ramos-Martínez, Heidegger N. Mateos-Toledo, Héctor Isaac Rocha-González, Juan Carlos Huerta-Cruz, Gloria Pérez-Rubio, Ingrid Fricke-Galindo, Jorge Rojas-Serrano, Ramcés Falfán-Valencia

**Affiliations:** 1Interstitial Lung Disease and Rheumatology Unit, Instituto Nacional de Enfermedades Respiratorias Ismael Cosío Villegas, Mexico City 14080, Mexico; daphne_935@outlook.com (D.R.-G.); medithmejia1965@gmail.com (M.M.); dr_heidegger@msn.com (H.N.M.-T.); 2Sección de Estudios de Posgrado e Investigación, Escuela Superior de Medicina, Instituto Politécnico Nacional, Mexico City 07700, Mexico; heisaac2013@hotmail.com; 3HLA Laboratory, Instituto Nacional de Enfermedades Respiratorias Ismael Cosío Villegas, Mexico City 14080, Mexico; krolnava@hotmail.com (K.J.N.-Q.); glofos@yahoo.com.mx (G.P.-R.); ingrid_fg@yahoo.com.mx (I.F.-G.); 4Unidad de Medicina Experimental, Facultad de Medicina, Universidad Nacional Autónoma de México, Mexico City 04510, Mexico; espiri77mx@yahoo.com; 5Laboratory of Clinical Pharmacology, Instituto Nacional de Enfermedades Respiratorias Ismael Cosío Villegas, Mexico City 14080, Mexico; pharman007@hotmail.com; 6Program of Masters and Ph.D. in Medical Sciences, School of Medicine, Universidad Nacional Autónoma de México, Mexico City 04510, Mexico

**Keywords:** antisynthetase syndrome, *MUC5B*, interstitial lung disease, usual interstitial pneumonia, genotype

## Abstract

**Background**: The presence of the rs35705950 variant in the *MUC5B* gene promoter is a critical genetic risk factor in idiopathic pulmonary fibrosis (IPF). It has been associated with usual interstitial pneumonia (UIP) in several interstitial lung diseases (ILDs). In antisynthetase syndrome (ASSD), most high-resolution computed tomography (HRCT) patterns are inflammatory, but up to 13% have UIP, leading to a worse prognosis. **Methods**: This single-center study included 60 patients with ASSD-ILD. We investigated whether carrying the *MUC5B* rs35705950 promoter variant was associated with UIP. To estimate the strength of the association between the genotype of the *MUC5B* rs35705950 promoter variant and the fibrotic pattern we used the odds ratio (cOR), and to assess the effect of confounding variables (age, evolution time, and sex), we performed a logistic regression to obtained the adjusted odds ratio (aOR). **Results**: The GT genotype of the *MUC5B* rs35705950 promoter variant is associated with up to a 4-fold increased risk of UIP (cOR 5.0, 95% CI 1.13–22.10), and the effect was even maintained after adjusting for potentially confounding variables such as sex, age, and time to progression (aOR 5.2, 95% CI 1.04–25.89). **Conclusions**: our study supports the role of *MUC5B* rs35705950 in ASSD-ILD with UIP. It reinforces that this polymorphism in our population could have a similar genetic basis to that already described in other ILDs that present predominantly fibrotic patterns.

## 1. Introduction

Antisynthetase syndrome (ASSD) is a rare multisystemic connective tissue disease with heterogeneous clinical manifestations, predominantly affecting the skin, joints, muscles, and lungs [1]. Aminoacyl transfer RNA synthetase (ARS) is a cytoplasmic protein that catalyzes the binding of amino acids to their corresponding transfer RNA in an energy-dependent manner [2]. In ASSD, the presence of anti-aminoacyl transfer-RNA-synthetases (anti-tRNA) autoantibodies has been well documented; these include anti-Jo1 (anti-histidyl), anti-PL12 (anti-alanyl), anti-PL7 (anti-threonyl), anti-EJ (anti-glycyl), anti-OJ (anti-isoleucyl), anti-SC (anti-lysil), anti-KS (anti-asparaginyl), anti-JS (anti-glutaminyl), anti-Ha (anti-tyrosyl) or anti-YRS (anti-threonyl), anti-tryptophanyl, and anti-Zo (anti-phenylalanyl), with anti-Jo1 being the most frequent, having a frequency of 68% in patients with ASSD, and 25% in patients with idiopathic inflammatory myopathies (IIM) [3].

On the other hand, the *MUC5B* rs35705950 gene promoter variant, a common gain-of-function single-nucleotide variant (SNV), leads to increased MUC5B protein expression and subsequent overproduction of mucin 5B in the distal airways. This SNV is a critical genetic risk factor for idiopathic pulmonary fibrosis (IPF) in up to 50% of patients [4,5]. Carrying the mutant allele (T), either heterozygous (GT) or homozygous (TT), confers the main risk factor for the development of pulmonary fibrosis [6].

Furthermore, it has been identified as a risk factor for the development of usual interstitial pneumonia (UIP) in many interstitial lung diseases (ILDs), such as chronic hypersensitivity pneumonitis (CHP), asbestosis, and rheumatoid arthritis-associated interstitial lung disease (RA-ILD) [6,7,8,9,10].

In the context of antisynthetase syndrome (ASSD), interstitial lung disease (ILD) has a prevalence ranging from 67% to 100%, with this condition being the primary determinant of prognosis and mortality [11]. Although most patients present with predominantly inflammatory patterns in HCRT, such as non-interstitial pneumonia (NSIP) or organizing pneumonia (OP) [12,13], up to 13% of patients predominantly present fibrotic patterns such as UIP [14], leading to a poorer prognosis and limited response to immunosuppressive therapy [15].

HRCT patterns in ILD are broadly classified into fibrotic and non-fibrotic subtypes; fibrotic patterns, particularly the UIP, have a worse prognosis. In IPF patients, the presence of a UIP pattern is associated with a significant increase in mortality at 2-year follow-up. Subpleural honeycombing and the extent of fibrotic changes are recognized as independent prognostic predictors of mortality in this population [16]. The prognostic implications regarding UIP patterns in patients with ASSD remain less well-defined due to the limited data available. However, evidence suggests that these patients with a UIP pattern have a worse prognosis due to increased progression of ILD compared to inflammatory HRCT patterns [17].

Despite the uncertain precise etiopathogenesis of ILD-ASSD, research indicates a complex interplay between antibody subtype, environmental factors, and genetic components [18]. Information on the genetic aspects linked to the development of interstitial fibrotic patterns in this condition is limited. It has been previously reported that the *MUC5B* rs35705950 gene promoter variant was not associated with an increased risk of developing ILD in patients with inflammatory myopathies [19]. Nevertheless, only a few studies have explored its association with the UIP pattern in patients with ASSD-ILD. A study conducted with a European population reported a lack of association between the *MUC5B* rs35705950 promoter variant and the risk of UIP presentation [20].

Given the variability in any population’s genetics, particularly in some Latin American Mestizo populations, which harbor varying degrees of Caucasian contribution, and interactions with environmental factors, which may exert epigenetic influences on clinical manifestations and even the type of ILD, this study aimed to investigate whether the *MUC5B* rs35705950 promoter variant is associated with the development of patterns predominantly fibrotic, such as UIP, in a population of Mexican patients with ASSD-ILD [21].

## 2. Materials and Methods

This single-center study was performed from 2022 to 2023 at the rheumatology clinic and the HLA Laboratory of the Instituto Nacional de Enfermedades Respiratorias Ismael Cosío Villegas (INER) in Mexico City. The local institutional review board approved the protocol under code C06-23. All participants signed informed consent forms before enrolling in the study.

### 2.1. Study Population

Patients aged 18 years or older, independent of sex, with a confirmed diagnosis of interstitial lung disease (ILD) by high-resolution computed tomography (HRCT) three months before enrollment and positivity for one of the following autoantibodies were included: anti-Jo1, anti-PL7, anti-PL12, anti-OJ, and anti-EJ. A single rheumatologist (DR-G) evaluated all patients to assess the presence of clinical manifestations, epidemiological variables, and pharmacological treatment. All patients had to have undergone pulmonary function tests three months before enrollment in the study.

### 2.2. Measurement of Autoantibodies

Anti-ARS antibody subtypes (anti-Jo1, anti-PL7, anti-PL12, anti-OJ, and anti-EJ) were identified using immunoblot strips from the EUROLINE: Myositis Profile 3 commercial panel from EUROIMMUN, which is based in Lübeck, Germany, according to the manufacturer’s instructions.

### 2.3. Blood Sampling and DNA Isolation

Peripheral blood samples from each participant were drawn by venipuncture of the forearm and collected in tubes with dipotassium ethylenediaminetetraacetic acid (K2 EDTA, 1.8 mg/mL, BD Vacutainer, Franklin Lakes, NJ, USA) as anticoagulant and centrifuged at 4500× *g* revolutions per minute (RPM) for 5 min to separate peripheral mononuclear blood cells (PMBC) in a layer. The plasma was separated using micropipettes and stored at 80 °C until assayed; meanwhile, DNA was extracted from PMBC using the commercial Blood DNA Preparation—Solution Kit (Jena Bioscience GmbH, Jena, Germany) and then hydrated in TE buffer (Ambion, Waltham, MA, USA), according to the supplier’s recommendations. The DNA’s integrity was verified after completing electrophoresis in 1% agarose gel. The gel was examined under a UV transilluminator, and bands were detected. These solution-based genomic DNA purification kits guarantee minimal DNA fragmentation and yield DNA sized up to 150 kb. The presence and quality of DNA in each sample were confirmed.

The DNA was quantified by ultraviolet/visible light spectrophotometry using a NanoDrop 2000 spectrophotometer device (Thermo Scientific, Wilmington, DE, USA), utilizing the ratios of 260/280 and 260/230 to evaluate the purity of each sample (1.8–2.0 and >2, respectively) and stored at −80 °C until further processing. Each sample was adjusted to 15 ng/µL for subsequent genotyping.

### 2.4. Genotyping of MUC5B Single Nucleotide Variants

A reaction mixture was prepared with the TaqMan probe and TaqMan Master Mix™ (Applied Biosystems, Foster City, CA, USA) and nuclease-free water. It was mixed and centrifuged at 1500× *g* RPM. The SNP was genotyped by real-time PCR employing a StepOnePlus™ Real-Time PCR System (Applied Biosystems, Carlsbad, CA, USA) by allelic discrimination through a predesigned TaqMan probe for the rs35705950 variant (C___1582254_20, Applied Biosystems, Carlsbad, CA, USA). The amplification reaction was performed in MicroAmp^®^ Optical 96-well reaction plates (Applied Biosystems; Woolston, UK), which included 3 µL of adjusted DNA per subject, following the supplier’s instructions. Four non-template controls (NTC) were included as negative controls; 5% of the samples were genotyped by duplicate-like allelic designation control. The thermal cycling settings were: denaturation at 60 °C for 30 s, followed by 40 cycles of 95 °C for 10 min and 95 °C for a 15 s alignment, and extension at 60 °C for 1 min and 4 °C and can be retired from the device. Genotype analysis was performed using TaqMan Genotyper v1.7.1 software (Applied Biosystems™ Real-Time PCR system, USA). Allele discrimination was conducted by the application SDS (sequence detection software) v. 1.4 (Applied Biosystems, San Francisco, CA, USA). This method is a reliable and widely used technique for genotyping single nucleotide variants.

### 2.5. Tomographic Assessment

HRCT was performed with 1.0- or 1.5-mm axial slices at 1 cm intervals and reconstructed using a high spatial frequency algorithm. A total of 20–25 HRCT images were acquired for each patient. An expert (M-M), who has a high interobserver agreement (intraclass correlation coefficient 0.90, 95% CI 0.84–0.94), evaluated the studies in a blinded manner and classified the HRCT pattern *according to the* IPF (an Update) and Progressive Pulmonary Fibrosis in Adults: An Official ATS/ERS/JRS/ALAT Clinical Practice Guideline, 2022. Fibrotic patterns on HRCT, such as UIP, were classified according to the presence of honeycombing with a subpleural distribution and basal predominance, with or without peripheral traction bronchiectasis or bronchiolectasis. Lung biopsies were not performed because these findings have a high positive predictive value, ranging from 90% to 100% [22,23]. Inflammatory patterns identified on HRCT included NSIP and OP, classified according to the definitions outlined in the Fleischner Society’s Glossary of Terms for HRCT [24]. The NSIP pattern was characterized by basal-dominated reticular abnormalities with traction bronchiectasis, peribronchovascular extension, and subpleural distribution, often associated with areas of ground-glass attenuation. The OP pattern was defined by the presence of bilateral patchy areas of consolidation, predominantly in the subpleural and lower lung regions [25].

### 2.6. Statistical Analysis

The statistical analysis was performed using Stata v. 14.2. Categorical variables are presented as frequencies and percentages. In contrast, depending on their distribution, numerical variables are presented as the mean ± standard deviation (SD) or median and interquartile range (IQR). The Shapiro–Wilk test was used to determine the distribution of the variables. The χ^2^ test was used to analyze the independence between nominal variables to estimate the strength of the association between the genotype of the *MUC5B* rs35705950 promoter variant and the fibrotic pattern we obtained’s crude odds ratio (cOR). Logistic regression analysis was performed to evaluate the effect of possible confounding variables (age, time of evolution, and sex) between the fibrotic pattern and genotype to obtain an adjusted odds ratio (aOR). All analyses were 2-tailed at a 95% confidence interval, and statistical significance was set at *p* ≤ 0.05.

## 3. Results

We included sixty patients with ILD who were positive for anti-ARS antibodies; the mean age was 58 ± 11 years, and 72% were women. Patients had the following frequencies of antibodies: 27% had anti-Jo1, 27% had anti-PL7, 28% had anti-PL12, 13% had anti-EJ, and 5% had anti-OJ. The median time from onset of respiratory symptoms to diagnosis of ASSD was 6 months. Respiratory symptoms were observed in 93% of patients, and 91.2% of the population had an overall HCRT extension by Goh index > 20%. Regarding musculoskeletal symptoms, 21.2% of the patients had muscle weakness, 22% had arthritis, and 54% presented with mechanic’s hands. The full description of the population has already been described [26].

Most of the tomographic patterns were inflammatory (85%, NSIP or OP), and 15% presented a fibrotic pattern (UIP). Concerning MUC5B rs35705950, 73% of patients had the GG genotype, 25% had the GT genotype, and only one had the TT genotype. The other characteristics are summarized in Table 1 and Appendix A.

The GT genotype of the MUC5B rs35705950 promoter variant was associated with up to a 4-fold increase in the UIP pattern compared to that of carriers of the GG genotype (Table 2). This effect was maintained even after adjusting for potential confounding variables such as sex, age, and time of evolution (*p* = 0.044, OR 5.2, 95% CI 1.04–25.89). (Table 3).

## 4. Discussion

Our main finding revealed an association between the rs35705950 promoter variant (G > T transversion) of the *MUC5B* gene and the UIP pattern. Carrying this polymorphism increases the probability of having UIP by up to four times. Consequently, the findings are meaningful and relevant because they may indicate a genetic background like that observed in conditions such as RA-ILD, CHP, and IPF in the Mexican population [8,9,27].

It has been reported that up to 24% of patients with ASSD have ILD progression despite immunosuppressive treatment, and risk factors associated with progression include low baseline FVC and especially the UIP pattern [17,28]. Even the antibody subgroup has been associated with greater severity and fibrotic extension of ILD but not with UIP [26], so being a carrier of this polymorphism plays an important role, suggesting a genetic predisposition for developing this pattern, which is associated with a worse prognosis.

Although no study has linked this polymorphism to the progression of ILD in ASSD, a longitudinal study in patients with CHP has shown that individuals with the GT/TT genotype have greater severity of ILD, as indicated by a more significant annual loss of predicted FVC% compared to those with the GG genotype, regardless of immunosuppressive treatments. In our cross-sectional study, we did not observe differences in pulmonary function tests (PFTs), which may be due to the lack of longitudinal follow-up to detect changes in lung function; another explanation is the small sample size of patients with the UIP pattern. However, we observed that 56% of GT genotype carriers in our patients had a UIP pattern, suggesting a possible increased risk of ILD progression associated with this fibrotic phenotype [7].

In contrast to our findings, a study conducted among individuals of European ancestry with ASSD revealed no difference between the UIP tomographic pattern and the genotype and allele frequencies of *MUC5B* rs35705950 among those with ASSD-ILD; this could be explained as being due to population genetic variability [20]. It has been reported that the frequency of the T allele is lower in the Mexican population. In a study of patients with RA-ILD, it was found that being a carrier of the *MUC5B* promoter variant was associated with a twofold increase in ILD among patients with RA, particularly among those with evidence of UIP. In addition, this study reported that the T allele frequency was lower than that in the United States of America and France (32.6, 28.8, and 16.4) [10]. These findings could explain the low frequency of the T allele in our study. The only patient carrying the TT genotype was an incident patient with ILD with less than one year of disease evolution, so we cannot rule out the possibility of this patient developing fibrosing patterns in the future, potentially UIP.

Patients with inflammatory patterns on HRCT have been reported to improve after six months of treatment. Nevertheless, approximately 13% do not respond [14]; this could be explained by a predominant fibrotic component of ILD. Early identification of these patients is crucial. Using genetic biomarkers to identify individuals at increased risk of developing a UIP pattern could support the implementation of therapeutic strategies that incorporate the early use of antifibrotic treatments. This reinforces that fibrotic lung patterns may be linked to this genetic basis, as observed in other interstitial lung diseases [29,30]. Second, it has crucial clinical implications. There is a poor response to immunosuppressive therapy and high mortality associated with fibrosis [31].

Our study is not exempt from limitations. First, our institution is a national referral center for patients with pulmonary conditions. Therefore, patients referred to our center may not necessarily represent the broader population of patients evaluated elsewhere. Second, the small sample size restricted our ability to include more covariates in the multivariate models. Third, we did not determine the plasma *MUC5B* protein or its relationship with the different genotypes of this SNV, so we cannot assess its levels and implications in patients with ASSD-ILD.

Despite these limitations, our study is critical because it could suggest a genetic basis different from that already reported in other countries [20], possibly because it is a Mestizo population.

## 5. Conclusions

In summary, our study supports the role of *MUC5B* rs35705950 in ASSD-ILD with a UIP pattern, emphasizing that fibrotic lung patterns are linked to a genetic basis with crucial clinical implications. Also, there is a poor response to immunosuppressive therapy and high mortality associated with fibrosis. Identifying ASSD-ILD patients carrying this variant allows us to identify patients at greater risk of developing UIP. This could enable close follow-up and, therefore, timely initiation of antifibrotic therapy, improving outcomes and reducing mortality rates in patients with ASSD-ILD. We consider it necessary to perform longitudinal studies with a larger sample and to determine the levels and expression of the protein encoded by this gene to assess its impact on the evolution of interstitial lung disease.

This study supports the role of *MUC5B* rs35705950 in ASSD-ILD with UIP. It reinforces that this polymorphism in our population could have a similar genetic basis to that already described in other ILDs that present predominantly fibrotic patterns.

## Figures and Tables

**Table 1 jcm-13-06159-t001:** Sociodemographic, tomographic, and respiratory functional test characteristics according to genotype in patients with antisynthetase syndrome.

Onset Characteristics	GG	GT	*p*-Value
	*n* = 44	*n* = 15	
Demographics			
Age, years	56.9 ± 10.3	61.5 ± 11.3	0.149
Sex, female ^§^	32 (72.7)	10 (66)	0.745
Tobacco smoking history ^§^	14 (31.8)	4 (26.6)	1.000
Time from respiratory symptoms to ILD diagnosis, months	5.5 (2–12)	12 (3–25)	0.367
HRTC pattern ^§^			
NSIP/OP	40(91)	10 (67)	0.038 *
UIP	4 (9)	5 (33)	
Antibody subtype ^§^			
Anti-Jo1	12 (27.0)	4 (26.6)	1.000
Non-anti-Jo1 (others)			
Anti-PL7	13 (29.5)	2 (13.3)	
Anti-PL12	10 (22.7)	7 (46.6)	
Anti-EJ	7 (15.9)	1 (6.66)	
Anti-OJ	2 (4.54)	1 (6.66)	
Pulmonary function tests			
FVC (%)	63.7 ± 23.9	70.2 ± 27.9	0.405
DL_CO_ (%)	52.8 ± 27.3	65.8 ± 26.2	0.140

Notes: Data are presented as numbers (percentages); otherwise, they are given as the mean ± standard deviation or median [interquartile range]. ILD: interstitial lung disease, NSIP: nonspecific interstitial pneumonia, UIP: usual interstitial pneumonia. ^§^ The chi-squared test was used for nominal and categorical variables. A parametric *t*-test was used to compare more than two groups. Otherwise, a nonparametric Mann–Whitney test was used. * Statistical significance at a *p*-value ≤ 0.05. The only patient in the TT group was excluded from the description, and their characteristics are described in the Appendix A.

**Table 2 jcm-13-06159-t002:** Genotypic and allelic frequencies of MUC5B rs35705950 in ASSD-ILD+ patients stratified according to the presence of UIP and non-UIP patterns.

*MUC5B* rs35705950	Usual Interstitial Pneumonia *n* = 9	Inflammatory Pattern *n* = 51	cOR	CI 95%	*p*-Value
Genotypes					
GG	4 (44%)	40 (78%)		-	Ref.
GT	5 (56%)	10 (20%)	5.0	1.13–22.10	0.038 *
TT	0	1 (2%)	-	-	-
Alleles					
G	13 (72%)	90 (88%)	2.852	0.67–10.65	0.082
T	5 (28%)	12 (12%)			

ASSD: antisynthetase syndrome; ILD: interstitial lung disease; OR: odds ratio; CI: confidence interval. Genotypes: GG: guanine–guanine; GT: guanine–thymine; TT: thymine–thymine. * Statistical significance at a *p*-value ≤ 0.05. The allele frequency analysis was calculated by dividing the number of times the allele of interest was observed in a population by the total number of copies of all the alleles [25].

**Table 3 jcm-13-06159-t003:** Associations between the fibrotic pattern and genotype in antisynthetase patients.

Variable	aOR	95% CI	*p*-Value
Genotype GT-GG	5.1059	1.012–25.759	0.048 *
Age, years	1.0590	0.972–1.154	0.191
Sex (ref. male)	1.6810	0.334–8.465	0.529
Time from respiratory symptoms to ILD diagnosis	0.9875	0.960–1.016	0.381

Adjusted logistic regression models. * Statistical significance at a *p*-value ≤ 0.05. Model 1: r^2^ = 0.222, constant = −5.684; aOR: adjusted odds ratio; 95% CI: 95% confidence interval.

## Data Availability

The data presented in this study are available upon request from the corresponding author. The data are not publicly available due to ethical reasons.

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
