# Peer review of "MUC5B rs35705950 Promoter Variant Is Associated with Usual Interstitial Pneumonia in Patients with Antisynthetase Syndrome"

_jcm, 2024, doi:10.3390/jcm13206159_

Round 1

Reviewer 1 Report

Comments and Suggestions for Authors

The authors have beautifully reported the importance of MUC5B in the progression of Interstitial Pneumonia and this gene can be used as a therapeutic target for the disease. I would recommed the authors to include a paragraph in the introduction section stating the current trend of therapy and medicines are now in the market to treat this disease. Authors should include that how drug drug interactions also lead to severe indecence of mortality in this diseaseand the importance of Ondasetron in this disesase help to reduce blood clotting

Authors should include these two citations which are very important 

1. Machine learning liver-injuring drug interactions with non-steroidal anti-inflammatory drugs (NSAIDs) from a retrospective electronic health record (EHR) cohort

A Datta, NR Flynn, DA Barnette, KF Woeltje, GP Miller… - PLoS computational biology, 

2. ‘Black box’to ‘conversational’machine learning: Ondansetron reduces risk of hospital-acquired venous thromboembolism

A Datta, MK Matlock, N Le Dang, T Moulin, KF Woeltje… - IEEE Journal of Biomedical and Health Informatics, 2020

Comments on the Quality of English Language

The authors have beautifully reported the importance of MUC5B in the progression of Interstitial Pneumonia and this gene can be used as a therapeutic target for the disease. I would recommed the authors to include a paragraph in the introduction section stating the current trend of therapy and medicines are now in the market to treat this disease. Authors should include that how drug drug interactions also lead to severe indecence of mortality in this diseaseand the importance of Ondasetron in this disesase help to reduce blood clotting

Authors should include these two citations which are very important 

1. Machine learning liver-injuring drug interactions with non-steroidal anti-inflammatory drugs (NSAIDs) from a retrospective electronic health record (EHR) cohort

A Datta, NR Flynn, DA Barnette, KF Woeltje, GP Miller… - PLoS computational biology, 

2. ‘Black box’to ‘conversational’machine learning: Ondansetron reduces risk of hospital-acquired venous thromboembolism

A Datta, MK Matlock, N Le Dang, T Moulin, KF Woeltje… - IEEE Journal of Biomedical and Health Informatics, 2020

Author Response

The authors have beautifully reported the importance of MUC5B in the progression of Interstitial Pneumonia and this gene can be used as a therapeutic target for the disease. I would recommend the authors to include a paragraph in the introduction section stating the current trend of therapy and medicines are now in the market to treat this disease. Authors should include that how drug interactions also lead to severe incidence of mortality in this disease and the importance of Ondasetron in this disease help to reduce blood clotting.

Authors should include these two citations which are very important:

  1. Machine learning liver-injuring drug interactions with non-steroidal anti-inflammatory drugs (NSAIDs) from a retrospective electronic health record (EHR) cohort

A Datta, NR Flynn, DA Barnette, KF Woeltje, GP Miller… - PLoS computational biology, 

  1. Black box’to ‘conversational’machine learning: Ondansetron reduces risk of hospital-acquired venous thromboembolism

A Datta, MK Matlock, N Le Dang, T Moulin, KF Woeltje… - IEEE Journal of Biomedical and Health Informatics, 2020.

  1. R. We appreciate the suggestions to cite the articles previously listed by the reviewer; we read them carefully. Both are really interesting papers; however, we consider that their subject matter is unrelated to our manuscript's main topic.

Reviewer 2 Report

Comments and Suggestions for Authors

This is an interesting study and the manuscript is well-written. Thus, I just have several minor suggestions.

1. Please add discussion about clinical implication

2. Pleaes move the last paragraph to the conclusion section.

Author Response

This is an interesting study, and the manuscript is well-written. Thus, I just have several minor suggestions.

  1. Please add discussion about clinical implication

  1. Thank you very much for this suggestion; the following paragraphs were added to the discussion emphasizing the clinical implications of our results:

“This finding has a significant clinical implication. Although most patients present with inflammatory patterns on HRCT and improve after six months of treatment, approxi-mately 13% do not respond [20], which could be explained by a predominant fibrotic component of ILD. Early identification of these patients is crucial. Using genetic biomarkers to identify individuals at increased risk of developing a UIP pattern could support the implementation of therapeutic strategies that incorporate the early use of antifibrotic treatments. This reinforces that fibrotic lung patterns may be linked to this genetic basis, as observed in other interstitial lung diseases [22, 23].” (Lines 222 to 229)

  1. Please move the last paragraph to the conclusion section.

  1. This valuable contribution has been considered, and the last paragraph has been moved to the conclusions section.

Reviewer 3 Report

Comments and Suggestions for Authors

The current study was aimed at identifying the association of MUC5B rs35705950 promoter variant with an increased risk of developing usual interstitial pneumonia (UIP) in patients with antisynthetase syndrome-associated interstitial lung disease (ASSD-ILD). The research finding in the current study seems consistent with the previous knowledge on the link between genetic variant and idiopathic pulmonary fibrosis (IPF) and other interstitial lung diseases (ILDs) with fibrotic patterns. The detailed statistical analysis ran on the data set obtained from the current study revealed an undisputed association between MUC5B variant and UIP against the other confounding factors like age, sex and disease progression time. The adjusted odds ratio (aOR 5.2) supports that the effect of this genetic variant on the risk of UIP remains significant. The current study provided a logical interpretation of data suggesting the potential importance of genetic screening for the MUC5B variant in patients with ASSD-ILD as a tool to screen for the prognostic information benefiting patients with fibrotic UIP patterns.

The single-center prospective study enrolled a cohort of adult (>18 yrs independent of sex), prescreened three months prior the study initiation for expression of one of the autoantibodies (Anti-aminoacyl-tRNA synthetase (anti-ARS) antibodies subtypes) anti-Jo1, anti-PL7, anti-PL12, anti-OJ, and anti-EJ. Clinical manifestations, epidemiological variables, and pharmacological treatment history were evaluated by one of the trained investigators. Necessary pulmonary 92 function tests were run on each patients prior three month of study initiation.  The study took advantage of techniques such as Blood sampling and DNA isolation, Genotyping of MUC5B single nucleotide variants, Tomographic assessment and performed necessary statistical data analysis for interpretation purposes. Based on the previous knowledge that MUC5B rs35705950 gene promoter variant, a common gain of-function single-nucleotide variant (SNV) leads to subsequent overproduction of mucin 5B in the distal airways and a possible risk factor for IPF in up to 50% of patients the heterozygous (GT) mutant alleles were analyzed in detail by a small cohort of patients (GT:n=15) and a cohort of normal GG genotype (GG:n=44) was used as a control comparator group.

Major:

1.       What was the total number of participants enrolled in the study? Was it 60 (GG:n=44, GT:n=15 and TT:n=1)? Is TT prevalence relatively low in IPF patients? What are the global prevalence of MUC5B rs35705950 SNV?  The TT genotype cohort consisted only one patient and data not included in the Table 1. Please present this data as supplementary.

2.       I would like to see the expression of other Non-anti-Jo1 antibodies in GG and GT cohorts with more granularity.

3.       Apart from the confirmed (G>T) mutation the functional FVC (%) and DLCO (%) do not seem to be any different between GG and TT cohorts. How this finding could be explained?

4.       Why the n values for GG and TT do not match with the values presented in table 2? Were they dropped for any reasons?

Minor:

1.       Please identify the significant p values in the tables 1-3 by using superscripts. Showing significance in bold face is not easy to identify.

Author Response

Major:

  1. What was the total number of participants enrolled in the study? Was it 60 (GG:n=44, GT:n=15 and TT:n=1)? Is TT prevalence relatively low in IPF patients? What are the global prevalence of MUC5B rs35705950 SNV?  The TT genotype cohort consisted only one patient and data not included in the Table 1. Please present this data as supplementary.

We acknowledge the relevance of your comment regarding the number of participants included in the study. Indeed, there were 60 patients with the following genotype distribution: GG (n=44), GT (n=15), and TT (n=1). We have included a supplementary table to present the three genotypes and their corresponding characteristics in response to your recommendation on describing the TT genotype.

Regarding your question, the prevalence of the MUC5B single nucleotide variant (SNV) rs35705950 has been reported as 41.9% in Caucasian patients with idiopathic pulmonary fibrosis (IPF) and 10.8% in the control group. In contrast, the T risk allele frequency in the Chinese population is approximately 3.33% in IPF patients and 0.66% in controls. In Mexico, the prevalence of this allele in healthy individuals, as well as in IPF and interstitial lung disease (ILD) associated with anti-synthetase syndrome (the pathology described in this article), remains unknown. However, the discussion section of this article references a recent study reporting a T allele frequency of 16.4% in ILD associated with rheumatoid arthritis. Notably, the prevalence in our population was the lowest compared to that reported in the USA (32.6%) and France (28.8%). We, therefore, consider that this low prevalence could explain the single TT genotype in our sample.

  1. Wu, X., Li, W., Luo, Z., & Chen, Y. (2021). The minor T allele of the MUC5B promoter rs35705950 associated with susceptibility to idiopathic pulmonary fibrosis: a meta-analysis. Scientific reports11(1), 24007. https://doi.org/10.1038/s41598-021-03533-z

  1. I would like to see the expression of other Non-anti-Jo1 antibodies in GG and GT cohorts with more granularity.

  1. R. Considering your relevant suggestion, the frequencies and percentages of the No-Jo1 subgroup were added to Table 1.

3.Apart from the confirmed (G>T) mutation the functional FVC (%) and DLCO (%) do not seem to be any different between GG and TT cohorts. How this finding could be explained?

  1. We thought it was very important to add information in the discussion about your suggestion about the difference in PFTs between both genotypes (GG vs. GT), so we added the following paragraph explaining the main reasons why we did not find differences in this cross-sectional study:

“In our cross-sectional study, we did not observe differences in pulmonary function tests (PFTs), which may be due to the lack of longitudinal follow-up to detect changes in lung function, or another explanation is the small sample size among patients with UIP pattern. “ (Lines 199 to 208)

  1. Why the n values for GG and TT do not match with the values presented in table 2? Were they dropped for any reasons?

Given the importance of your suggestion, we have clarified the information in the table related to allele analysis by adding a description of the method used to obtain the frequencies, along with a reference that explains the methodology in detail.

As follows:
“The allele frequency analysis was calculated by dividing the number of times the allele of interest is observed in a population by the total number of copies of all the alleles."

1.Kim, S.Y., Lohmueller, K.E., Albrechtsen, A. et al. Estimation of allele frequency and association mapping using next generation sequencing data. BMC Bioinformatics 12, 231 (2011). https://doi.org/10.1186/1471-2105-12-231

Minor: 

  1. Please identify the significant p values in the tables 1-3 by using superscripts. Showing significance in bold face is not easy to identify.

  1. The suggestion has been considered, and a superscript (*) has been added to show statistical significance in all tables (p ≤ 0.05).

Round 2

Reviewer 3 Report

Comments and Suggestions for Authors

The authors have adequately addressed my comments and concerns and I am satisfied with the responses. The revised manuscript appears much improved.